# Trends in heart failure-related cardiovascular mortality in rural versus urban United States counties, 2011–2018: A cross-sectional study

**Jacob B. Pierce**[1], **Nilay S. Shah**[2], **Lucia C. Petito**[3], **Lindsay Pool**[3], **Donald M. Lloyd-Jones**[2,3], **Joe Feinglass**[3,4], **Sadiya S. Khan**[2,3]*

**1** Feinberg School of Medicine, Northwestern University, Chicago, Illinois, United States of America, **2** Division of Cardiology, Department of Medicine, Feinberg School of Medicine, Northwestern University, Chicago, Illinois, United States of America, **3** Department of Preventive Medicine, Feinberg School of Medicine, Northwestern University, Chicago, Illinois, United States of America, **4** Division of General Internal Medicine and Geriatrics, Department of Medicine, Northwestern University Feinberg School of Medicine, Chicago, Illinois, United States of America

* s-khan-1@northwestern.edu

**Data Availability Statement:** The data underlying the results presented in the study are available from CDC WONDER (https://wonder.cdc.gov/mcd.

## Abstract

### Background

Adults in rural counties in the United States (US) experience higher rates broadly of cardiovascular disease (CVD) compared with adults in urban counties. Mortality rates specifically due to heart failure (HF) have increased since 2011, but estimates of heterogeneity at the county-level in HF-related mortality have not been produced. The objectives of this study were 1) to quantify nationwide trends by rural-urban designation and 2) examine county-level factors associated with rural-urban differences in HF-related mortality rates.

### Methods and findings

We queried CDC WONDER to identify HF deaths between 2011–2018 defined as CVD (I00-78) as the underlying cause of death and HF (I50) as a contributing cause of death. First, we calculated national age-adjusted mortality rates (AAMR) and examined trends stratified by rural-urban status (defined using 2013 NCHS Urban-Rural Classification Scheme), age (35–64 and 65–84 years), and race-sex subgroups per year. Second, we combined all deaths from 2011–2018 and estimated incidence rate ratios (IRR) in HF-related mortality for rural versus urban counties using multivariable negative binomial regression models with adjustment for demographic and socioeconomic characteristics, risk factor prevalence, and physician density. Between 2011–2018, 162,314 and 580,305 HF-related deaths occurred in rural and urban counties, respectively. AAMRs were consistently higher for residents in rural compared with urban counties (73.2 [95% CI: 72.2–74.2] vs. 57.2 [56.8–57.6] in 2018, respectively). The highest AAMR was observed in rural Black men (131.1 [123.3–138.9] in 2018) with greatest increases in HF-related mortality in those 35–64 years (+6.1%/year). The rural-urban IRR persisted among both younger (1.10 [1.04–1.16]) and older adults (1.04 [1.02–1.07]) after adjustment for county-level factors. Main limitations included lack of individual-level data and county dropout due to low event rates (<20).

html), US Census Bureau Population and Housing Unit Estimates (https://www.census.gov/programs-surveys/popest.html), Small Area Income and Poverty Estimates (SAIPE) Program (https://www.census.gov/programs-surveys/saipe.html), Small Area Health Insurance Estimates (SAHIE) Program (https://www.census.gov/programs-surveys/sahie/data.html), US Bureau of Labor Statistics Local Area Unemployment (https://www.bls.gov/lau/), Health Resources and Services Administration Area Health Resources File (https://data.hrsa.gov/topics/health-workforce/ahrf), and CDC Diabetes Surveillance System (https://gis.cdc.gov/grasp/diabetes/diabetesatlas.html#). The years used in our analyses are detailed in S1 Table.

**Funding:** This work was supported by grants from National Institutes of Health's National Center for Advancing Translational Sciences Grant Number KL2TR001424 and the American Heart Association (#19TPA34890060) to SSK including salary support. The funders had no role in study design, data collection and analysis, decision to publish, or preparation of the manuscript.

**Competing interests:** The authors have declared that no competing interests exist.

**Abbreviations:** AAMR, age-adjusted mortality rate; AAPC, average annual percent change; CDC WONDER, Center for Disease Control and Prevention Wide-Ranging Online Data for Epidemiologic Research; CVD, cardiovascular disease; HF, heart failure; IRR, incidence rate ratio; PCP, primary care physician.

## Conclusions

Differences in county-level factors may account for a significant amount of the observed variation in HF-related mortality between rural and urban counties. Efforts to reduce the rural-urban disparity in HF-related mortality rates will likely require diverse public health and clinical interventions targeting the underlying causes of this disparity.

## Introduction

The excess burden of cardiovascular mortality experienced by rural communities in the United States (US) has recently come to the forefront of national conversations on health disparities, highlighted by the recent American Heart Association Call to Action on Rural Health [1]. Individuals living in rural communities face significant challenges in achieving optimal cardiovascular health, contributing to well-known rural-urban differences in overall mortality and cardiovascular disease (CVD) burden [1–3]. Importantly, rural-urban disparities have continued to widen in recent years, and rural communities now have over a 20% higher all-cause mortality rate than their urban counterparts [1].

As declines in the CVD mortality rate in the US have stalled, heart failure (HF)-related mortality has begun to increase since 2011 [4–6], likely as a result of increasing prevalence of risk factors for HF-related mortality [7–10] concurrent with poor implementation of guideline-directed medical therapy and novel therapeutic agents [11, 12]. Several recent studies have demonstrated significant heterogeneity in how age and race-sex subgroups have been affected by these shifting HF-related mortality rates [13, 14]. However, differences in recent HF-related mortality rates between rural and urban adults have not been described, and how rates may differ across age and race-sex subgroups in rural versus urban adults is unknown. Identifying county-level factors that are associated with differences in rural and urban HF-related mortality rates may help to identify potential targets for policy interventions.

Thus, the aim of our study was two-fold: first, to quantify national trends in HF-related mortality by conducting a nationwide analysis of HF-related deaths between 2011 and 2018 by rural-urban status overall and stratified by age and race-sex subgroups; and second, to examine the association between county-level factors (demographic and socioeconomic characteristics, risk factor prevalence, and physician density) and excess HF-related mortality in rural compared with urban counties.

## Materials and methods

### Study setting and population

This study used a serial cross-sectional design in which data were examined to determine the number of HF-related deaths in each county in the US annually between 2011 and 2018. A pre-specified analytic plan was finalized on February 11, 2020. Statistical analysis occurred between February 12 through March 27, 2020. We used the Centers for Disease Control and Prevention Wide-Ranging Online Data for Epidemiologic Research Multiple Cause of Death Online Database (CDC WONDER), which captures mortality stratified by age, race, sex, and county [15]. This dataset includes cause of death from death certificates for the 50 states and the District of Columbia. Similar to previous studies [16–20], data on county-level factors were obtained from the US Census Bureau Population and Housing Unit Estimates [21], Small Area Income and Poverty Estimates Program [22], Small Area Health Insurance Estimates

Program files [23]; US Bureau of Labor Statistics Local Area Unemployment Statistics file [24]; Centers for Disease Control and Prevention Behavioral Risk Factor Surveillance System [25]; and the Health Resources and Services Administration Area Health Resources File [26] and subsequently linked using county identifiers to the CDC WONDER mortality data. The rationale for our analytic approach is based on the recommendations by the United States Centers for Disease Control (CDC) and National Vital Statistics System (NVSS) who curate and provide access to these nationwide data and is consistent with prior CDC publications [27–30]. This study was exempt from institutional review board review at Northwestern University Feinberg School of Medicine due to the publicly available, deidentified nature of the data and follows the Strengthening the Reporting of Observational Studies in Epidemiology (STROBE) guidelines for reporting (S1 Checklist).

## U.S. HF-related deaths: CDC WONDER 2011–2018

We obtained data on all HF-related deaths among Black and White US adults aged 35 to 84 between January 1, 2011 and December 31, 2018. The study period began at the well-described inflection point in 2011 at which point HF-related mortality in the United States began to increase [4–6] and extends through the most recent year for which data are publicly available in CDC WONDER, 2018. As HF is generally considered to be a mediator between disease and death rather than an underlying cause of death by nosologists, coding instructions suggest that other diseases be considered the underlying cause of death. Therefore, to capture the broad burden of cardiovascular mortality related to HF, we included decedents in which HF (defined as *International Classification of Diseases, 10th Revision* [ICD-10] code I50) was listed as a multiple cause of death with CVD (I00-I78) listed as the underlying cause of death similar to previous studies [10, 14, 31].

We additionally extracted data on age, race, sex, and county from CDC WONDER to characterize individuals who experienced a HF-related death. Race was classified as Black or White. Age was categorized as 35–64 years and 65–84 years based on age of 65 years being required for Medicare eligibility and previously published analyses [5, 10, 14]. Decedents ages 85 years and older were excluded due to low rates of reliability for coding related to cardiovascular causes of mortality in this age group [32]. The subsequent younger and older age group stratifications were chosen on the basis of recent increases in observed cardiovascular-specific HF-related mortality among adults age 35–64 since 2011 [14, 33]. Due to concerns over the reliability of accurate reporting of ethnicity (i.e. Hispanic or Latinx) [15] on death certificate data, we stratified only by race (Black or White), for which death certificates have been shown to be exceedingly accurate [34]. Too few HF-related mortality events precluded the inclusion of other race and ethnicity groups (e.g. Native American, Asian, or Pacific Islander). We utilized the 2013 NCHS Urban-Rural Classification Scheme for Counties [35] to classify counties as rural (micropolitan and noncore) or urban (large central metro, large fringe metro, medium metro, and small metro) as in prior published analyses. [36–39].

## County-level factors

Publicly available county-level data were obtained regarding percentage of female, non-Hispanic Black, and Hispanic residents; percentage of residents in poverty (household income below the federal poverty threshold), unemployed, and uninsured (age 18–64 years); median household income of residents; percentage of residents with diabetes and obesity; and number of primary care physicians and cardiologists per 100,000 residents. The sources, files, and years used for primary and sensitivity analyses are detailed in S1 Table.

## Statistical analysis

To examine nationwide trends in HF-related mortality, we calculated annual crude and age-adjusted mortality rates (AAMRs) from 2011 to 2018 by rural-urban status for the overall study population as well as in each age and race-sex subgroup. Mortality rates were age-adjusted using the direct method with the 2000 US census as the standard population [40]. To quantify nationwide annual trends in HF-related mortality, we calculated the average annual percent change (AAPC) in AAMR between 2011 and 2018 using the Joinpoint Regression Program version 4.7 (National Cancer Institute). In the nationwide trend analyses, decedents were stratified by rural-urban status, and trends were examined overall and for age and race-sex subgroups.

We further calculated individual county-level AAMRs from all HF-related deaths that occurred over the entire 8-year study duration. Deaths were combined over the study period in order to maximize county inclusion, as counties with <20 cumulative events were censored for confidentiality purposes or "unreliable" reported AAMRs by CDC WONDER. Small numbers of HF-related mortality events at the county level precluded stratifying by race-sex subgroups in the county-level analysis. We then investigated geospatial differences in HF-related mortality by mapping US counties according to their quintile of HF-related AAMR. Mapping was performed using ArcGIS version 10.6.1 (Esri, Redlands, CA).

In a cross-sectional county-level analysis, we investigated whether individual county-level factors were associated with overall HF-related mortality rates in 2011–2018. We used multivariable negative binomial regression models to estimate incidence rate ratios (IRRs) for HF-related mortality in rural compared with urban counties stratified by age of decedent. Due to overdispersion of the data used in our regression analyses, we utilized negative binomial regression rather than Poisson regression. We first modeled the unadjusted association between rural status and county-level AAMR (Model 1). Models 2–5 adjusted for individual county-level covariate categories, including demographic characteristics (Model 2; percent of residents over age 65 years, percent of female residents, percent of non-Hispanic Black residents, and percent of Hispanic residents), socioeconomic characteristics (Model 3; percent of residents in poverty, percent of residents unemployed, percent of uninsured residents age 18–64, and median household income), prevalence of clinical risk factors (Model 4; percent of residents with self-reported diabetes and obesity), and physician density (Model 5; number of PCPs and number of cardiologists per 100,000 residents). Model 6 adjusted for all covariates included in models 1–5 in one model. We also investigated the association of each individual county-level factor with HF-related mortality in the fully adjusted model by stratifying counties by quintiles of each factor and examining the IRR for HF-related mortality relative to the lowest quintile of each county-level factor. Our modeling strategy was designed to provide insight into the association of both broad categories of covariates (i.e. demographic characteristics, socioeconomic characteristics, risk factor prevalence, and physician density) and individual county-level covariates (e.g. percentage of county residents who are uninsured) to rural-urban disparities in HF-related mortality. Individual county-level covariates were chosen based upon known association with HF and cardiovascular disease morbidity and mortality with reliable, publicly available county-level data and consistent with prior publications examining differences in county-level mortality broadly in heart disease [16].

Finally, we conducted sensitivity analyses in which we repeated negative binomial regression modeling using the most recent publicly available county-level factors (S1 Table). STATA/IC software version 15.1 (College Station, TX) was used for all statistical analyses. Two-tailed $P < 0.05$ was considered statistically significant.

## Results

### Nationwide trends in HF-related mortality rates for rural versus urban adults, 2011–2018

Between 2011 and 2018, 742,619 HF-related deaths occurred among adults age 35 to 84 years in the US, including 162,314 rural and 580,305 urban deaths (S2 Table). In every year, HF-related AAMRs were higher for rural compared with urban adults (e.g., 73.2 [95% confidence interval (CI): 72.2–74.2] versus 57.2 [95% CI: 56.8–57.6] deaths per 100,000 residents in 2018). Both rural and urban adults experienced increases in annual HF-related mortality between 2011 and 2018 (AAPC = +1.3% [0.9–1.8%]. and +1.2% [0.7–1.7%] for rural and urban, respectively; Fig 1, Table 1). Regardless of rural-urban status, a greater relative annual increase in HF-related mortality rates occurred for younger (AAPC = +4.6% [95% CI: 3.7–5.5%] and +4.4 [95% CI: 4.0–4.9%] for rural and urban, respectively) compared with older adults (AAPC = +1.3% [95% CI: 0.9–1.8%] and +1.2 [95% CI: 0.7–1.7%] for rural and urban, respectively).

After stratifying by rural-urban status, race-sex subgroups also demonstrated differences in HF-related mortality trends (Table 1). Among young rural decedents in 2018, the AAMR among Black women (32.7 [95% CI: 28.8–36.7]) was 3.0-fold higher than that of rural White women (11.0 [95% CI: 10.3–11.6]), and the AAMR of Black men (52.3 [95% CI: 47.4–57.3]) was 2.5-fold higher than that of White men (20.8 [95% CI: 19.8–21.7]). Young, rural Black men had the largest AAPC in HF-related mortality over the study period at an average of +6.1% (95% CI: 3.7 to 8.5%) per year. Similar disparities existed between race-sex subgroups in young urban adults. Older urban White women were the only subgroup that did not experience a significant increase in HF-related mortality over the study period.

### County-level geospatial heterogeneity for HF-related AAMR, 2011–2018

Geospatial analysis of combined county-level HF-related deaths from 2011 to 2018 demonstrated marked county-level variation in HF-related mortality rates (Fig 2). Counties at the 90th percentiles had a 2.3-fold higher AAMR compared with counties at the 10th percentile (AAMR = 100.3 and 43.6 per 100,000 population for 90th and 10th percentiles, respectively). Counties in the highest quintile of HF-related AAMR (highest rates of HF-related mortality)

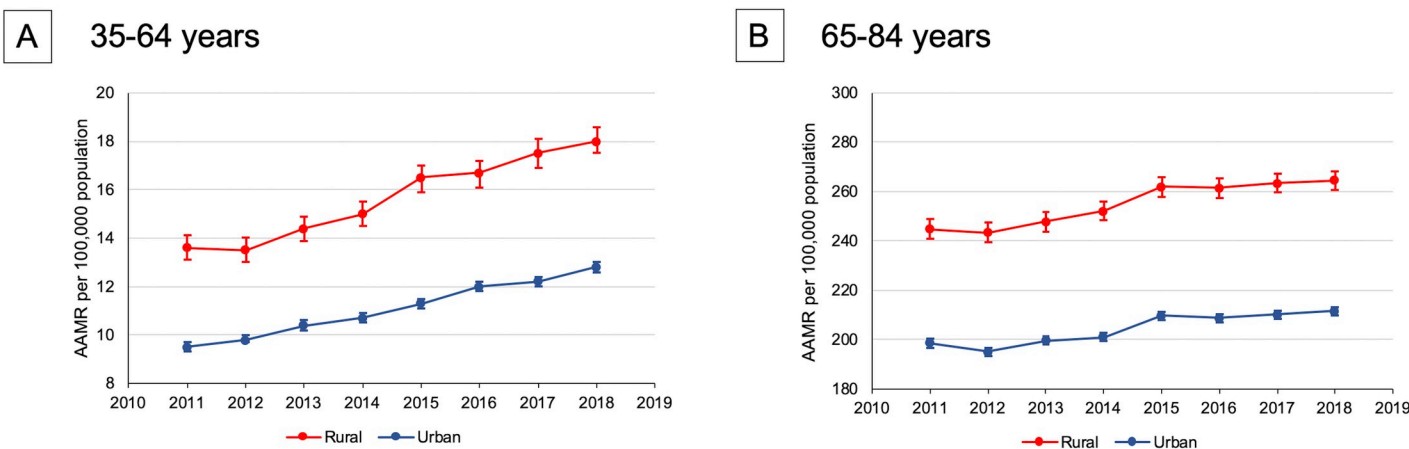

**Fig 1. Annual nationwide heart failure-related mortality rates stratified by age and rural-urban status, Centers for Disease Control and Prevention Wide-Ranging Online Data for Epidemiologic Research 2011–2018.** Heart failure mortality in the United States between 2011–2018 according to rural-urban status among adult age (A) 35–64 years and (B) 65–84 years. Rural-urban status determined based on the 2013 NCHS Urban-Rural Classification Scheme for Counties. AAMR expressed as number of deaths per 100,000 population. AAMR = age-adjusted mortality rate.

**Table 1. Age-adjusted mortality rates and average annual change in heart failure-related mortality between rural versus urban adults, overall and stratified by age and race-sex subgroups, center for disease control and prevention wide-ranging online data for epidemiologic research 2011–2018.**

| | Overall | | | Age 35–64 y | | | Age 65–84 y | | |
|---|---|---|---|---|---|---|---|---|---|
| | AAMR 2011 (per 100,000 population)* | AAMR 2018 (per 100,000 population)* | AAPC† | AAMR 2011 (per 100,000 population)* | AAMR 2018 (per 100,000 population)* | AAPC† | AAMR 2011 (per 100,000 population)* | AAMR 2018 (per 100,000 population)* | AAPC† |
| **Rural‡** | | | | | | | | | |
| Overall | 65.3 (64.3–66.3) | 73.2 (72.2–74.2) | +1.9 (1.4–2.4) | 13.6 (13.1–14.1) | 18.0 (17.5–18.6) | +4.6 (3.7–5.5) | 244.7 (240.8–248.7) | 264.5 (260.7–268.3) | +1.3 (0.9–1.8) |
| Black women | 80.5 (74.8–86.3) | 88.7 (83.0–94.5) | +1.9 (0.8–3.1) | 27.0 (23.5–30.5) | 32.7 (28.8–36.7) | +4.2 (1.6–6.7) | 266.3 (243.7–288.9) | 283.1 (261.5–304.7) | +1.0 (0.2–1.9 |
| White Women | 51.4 (50.2–52.5) | 55.7 (54.5–56.9) | +1.3 (0.6–2.0) | 8.5 (7.9–9.0) | 11.0 (10.3–11.6) | +4.1 (2.6–5.6) | 200.2 (195.2–205.1) | 210.9 (206.1–215.7) | +0.9 (0.2–1.5) |
| Black men | 106.4 (98.5–114.2) | 131.1 (123.3–138.9) | +3.2 (1.9–4.5) | 35.9 (31.8–40.0) | 52.3 (47.4–57.3) | +6.1 (3.7–8.5) | 350.8 (318.7–382.8) | 404.4 (374.2–434.7) | +2.0 (1.1–2.9) |
| White men | 77.2 (75.6–78.8) | 86.8 (85.3–88.4) | +2.0 (1.5–2.5) | 15.6 (14.8–16.4) | 20.8 (19.8–21.7) | +4.5 (3.7–5.4) | 291.0 (284.5–297.6) | 316.1 (309.8–322.4) | +1.5 (1.1–2.0) |
| **Urban‡** | | | | | | | | | |
| Overall | 51.8 (51.4–52.2) | 57.2 (56.8–57.6) | +1.7 (1.2–2.1) | 9.5 (9.3–9.7) | 12.8 (12.6–13.0) | +4.4 (4.0–4.9) | 198.5 (196.8–200.2) | 211.4 (209.8–213.0) | +1.2 (0.7–1.7) |
| Black women | 56.2 (54.5–57.9) | 64.6 (63.0–66.1) | +2.3 (1.7–2.9) | 15.4 (14.6–16.3) | 19.8 (18.9–20.7) | +3.6 (2.7–4.5) | 197.7 (190.8–204.5) | 219.9 (213.6–226.3) | +1.9 (1.3–2.6) |
| White Women | 39.5 (39.0–40.0) | 40.5 (40.0–41.0) | +0.7 (0.1–1.2) | 5.2 (5.0–5.4) | 6.4 (6.2–6.6) | +3.4 (2.4–4.4) | 158.6 (156.5–160.7) | 159.0 (157.0–160.9) | +0.3 (-0.3–1.0) |
| Black men | 80.9 (78.4–83.3) | 101.9 (99.5–104.2) | +3.3 (2.6–3.9) | 27.1 (25.9–28.3) | 37.7 (36.4–39.1) | +4.8 (3.8–5.7) | 267.4 (257.4–277.3) | 324.3 (314.9–333.8) | +2.7 (1.9–3.5) |
| White men | 62.0 (61.3–62.7) | 69.0 (68.4–69.7) | +1.8 (1.4–2.1) | 10.2 (9.9–10.5) | 13.8 (13.5–14.1) | +4.6 (4.2–4.9) | 241.8 (238.8–244.8) | 260.7 (257.9–263.6) | +1.3 (0.9–1.8) |

AAMR = age-adjusted mortality rate; AAPC = average annual percent change; y = year

*AAMR per 100,000 calculated by direct method using 2000 US Census as the standard population

†Average annual percent change calculated using Joinpoint Regression Program version 4.7 (National Cancer Institute)

‡Rural-urban status grouped based on the 2013 NCHS Urban-Rural Classification Scheme for Counties.

were predominantly rural counties in the South whereas counties in the lowest quintile (lowest rates) were predominantly urban counties located in the Northeast and Florida peninsula. The West census region demonstrated significant geospatial heterogeneity in HF-related mortality rates regardless of rural-urban status.

## County-level factors associated with excess HF-related mortality in rural versus urban counties

Descriptions of socioeconomic and demographic factors, risk factor prevalence, and physician density in rural and urban counties included in the cross-sectional county-level analyses are detailed in S3 Table. Of note, there were no practicing cardiologists in 51.1% and 74.9% of

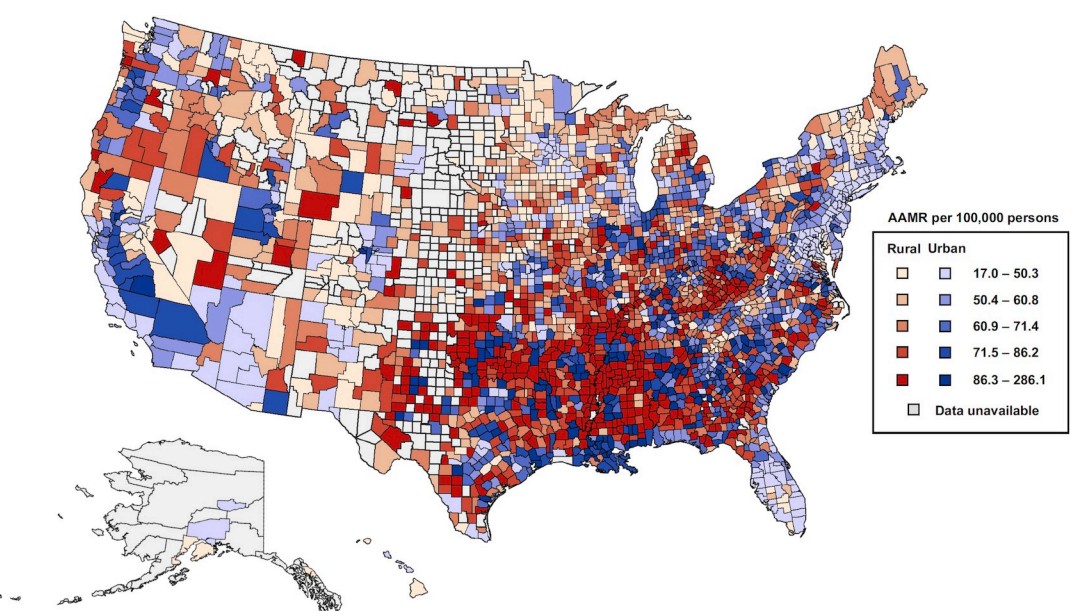

**Fig 2. County-level heart failure-related age-adjusted mortality rates in the United States, Centers for Disease Control and Prevention Wide-Ranging Online Data for Epidemiologic Research 2011–2018.** Counties were grouped according to their quintile of HF AAMR using pooled HF events between 2011 through 2018. Black and White decedents ages 35–84 were included. Counties censored for confidentiality purposes or counties with an AAMR reported as "unreliable" by CDC WONDER (<20 events) were excluded. AAMR = Age-adjusted mortality rate. Basemap Sources: Esri, TomTom North America, Inc., U.S. Census Bureau, U.S. Department of Agriculture (USDA), National Agricultural Statistics Service (NASS).

rural counties included in the county-level analyses of younger and older decedents, respectively. Unadjusted negative binomial regression models using individual county-level HF-related AAMRs showed that rural counties experienced significantly higher HF-related mortality rates compared with urban counties (IRR = 1.67 [95% CI: 1.57–1.78] and 1.16 [95% CI: 1.13–1.18] for younger and older adults, respectively; **Fig 3**). In separate regression models, the IRR for HF-related mortality in rural versus urban counties was significantly lower after separately adjusting for socioeconomic factors (IRR = 1.08 [95% CI: 1.03–1.14] and 1.01 [95% CI: 0.99–1.04] among younger and older adults, respectively) or risk factor prevalence (IRR = 1.25 [95% CI: 1.18–1.31] and 1.08 [95% CI: 1.06–1.10] among younger and older adults, respectively). Adjustment for demographic factors or physician density did not result in significant changes to the IRR. In fully-adjusted models for both younger and older adults, rural counties demonstrated moderately higher HF-related mortality when compared with urban counties (IRR = 1.10 [95% CI: 1.04–1.16] and 1.04 [95% CI: 1.02–1.07] for younger and older adults, respectively). In a sensitivity analysis using county-level factors from the end of the study period compared with the beginning of the study period (2016–2018 vs. 2010–2011), the association between rural status and HF-related mortality was largely unchanged (S4 Table).

Within each category, AAMRs varied across quintiles of the individual county-level covariates in the fully adjusted model with significant differences between urban and rural counties (**Fig 4**, S5 Table). Socioeconomic characteristics were consistently associated with differences in HF-related mortality, particularly median household income and the percent of residents who were uninsured. Higher density of PCPs was associated with significantly lower HF-related mortality rates in younger adults in rural counties, and higher density of cardiologists was associated with lower HF-related mortality rates for younger adults regardless of rural-urban status.

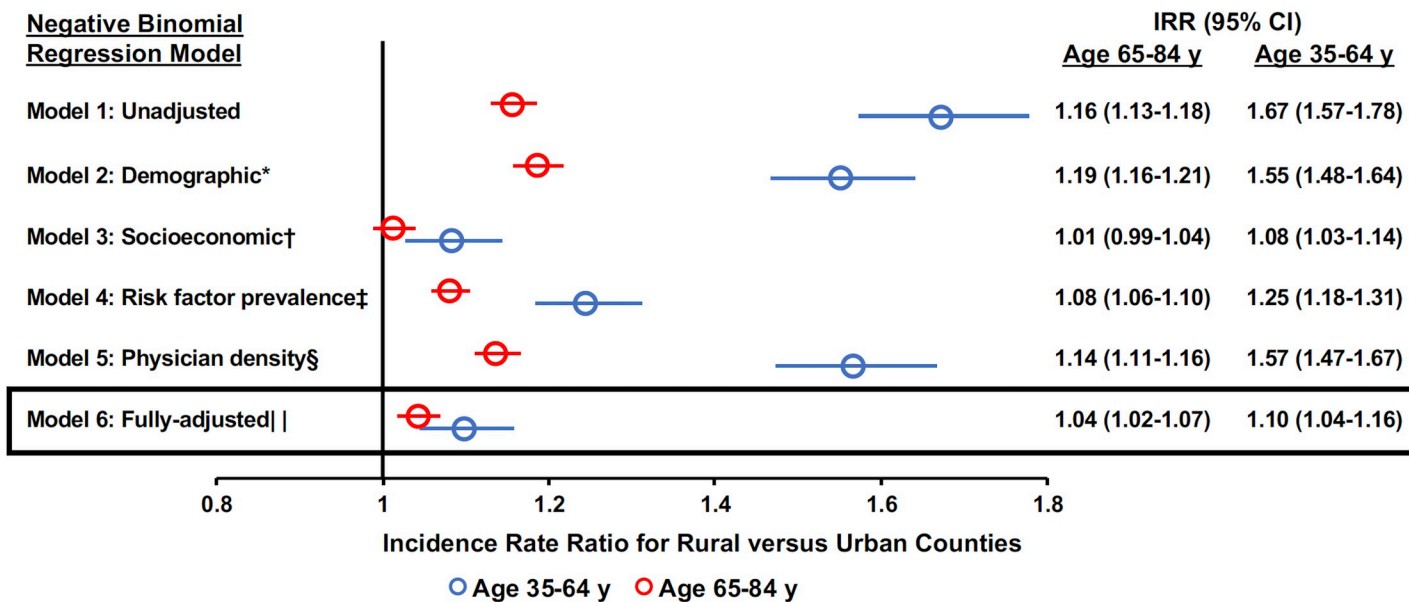

**Fig 3. Contributions of aggregate county-level factors to the rural heart failure-related mortality penalty.** Forest plots for negative binomial regression models of rural status and HF-related mortality adjusted for groups of county-level factors. *Adjusted demographic factors including percent of residents over age 65 years, percent of female residents, percent of non-Hispanic Black residents, and percent of Hispanic residents according to the US Census Bureau 2011 Population and Housing Unit Estimates. †Adjusted for socioeconomic factors including percent of residents in poverty, percent of residents unemployed, percent of residents uninsured age 18–64, and median household income according to the US Census Bureau 2011 Small Area Income and Poverty Estimates Program and Small Area Health Insurance Estimates Program. ‡Adjusted for clinical characteristics of residents including percent of residents with diabetes and percent of residents with obesity from the Centers for Disease Control and Prevention 2011 Behavioral Risk Factor Surveillance System. §Adjusted for clinician density including number of primary care physicians and number of cardiologists per 100,000 residents according to the Health Resources and Services Administration Area Health Resources File (primary care physician density from 2011 and cardiologist density from 2010). | |Adjusted for all covariates in models 2–5.

## Discussion

### Principal findings

In this nationwide study, we found marked county-level variation in HF-related mortality rates across the US with a significantly greater burden in rural counties, especially those in the South. HF-related mortality rates increased in both rural and urban counties with greater increases in HF-related mortality among younger compared with older adults between 2011 and 2018. Models accounting for differences in county-level factors significantly attenuated the excess in HF-related mortality rates in rural versus urban counties.

Our data extend and expand upon prior reports by highlighting growing disparities in HF-related mortality rates between rural and urban areas. The geographic distribution of HF-related mortality rates in the US presented here is consistent with previous reports of higher HF hospitalization rates and greater prevalence of HF risk factors such as diabetes and obesity in the South [41, 42]. States in the South have consistently had higher overall CVD mortality rates compared with the rest of the country [43, 44]. Our data extend these previous analyses by demonstrating significant county-level heterogeneity in HF-related mortality rates, identifying the need to focus on the community level for prevention.

This marked county-level variation prompted us to identify key county-level factors that appear to contribute to the observed heterogeneity in HF-related mortality rates. Adjustment for county-level socioeconomic status resulted in the largest attenuation in excess HF-related mortality rates in rural compared with urban counties. These results are in agreement with previous studies demonstrating the important relationship between county-level

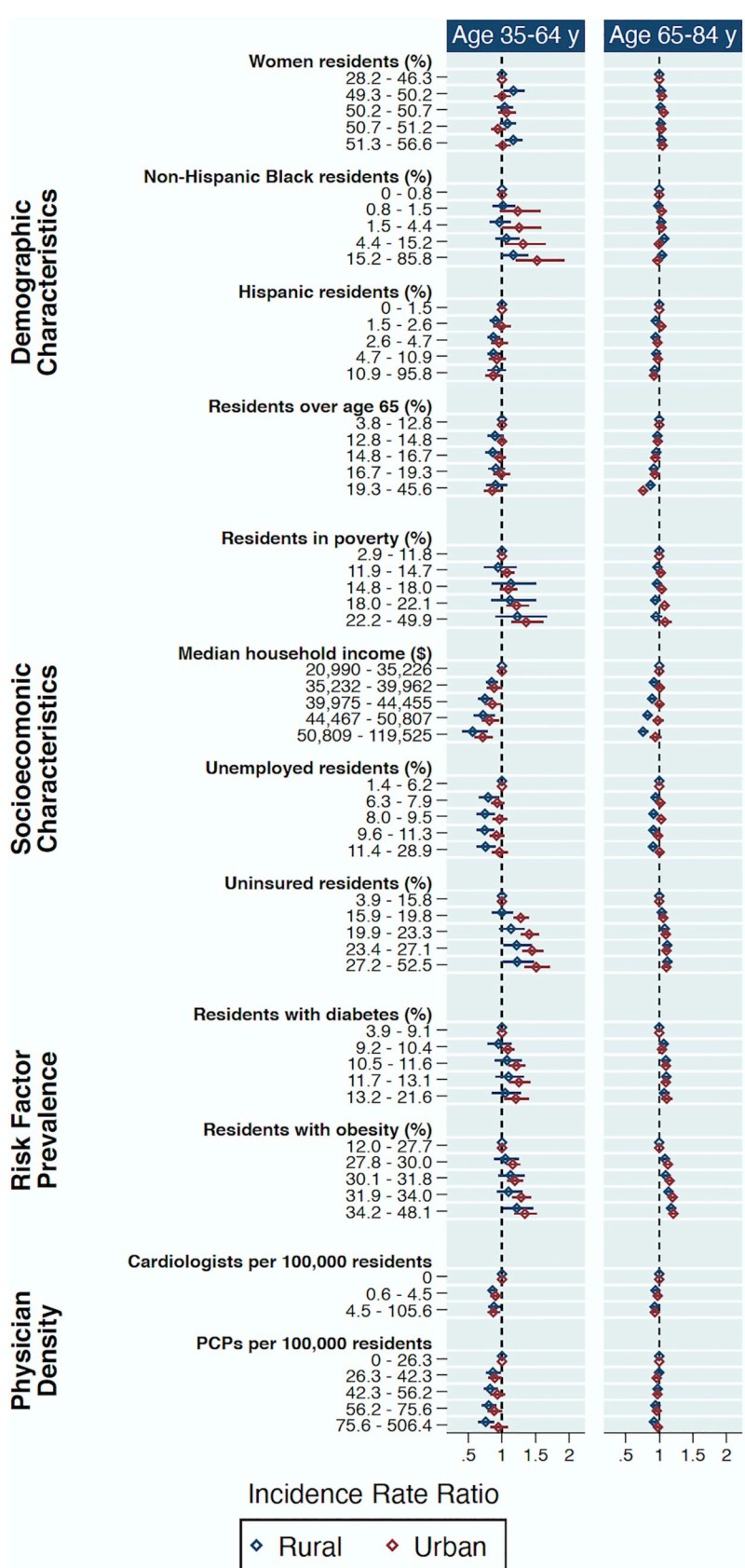

**Fig 4. Associations between county-level factors and age-adjusted heart failure-related mortality rates stratified by rural-urban status and age, Centers for Disease Control and Prevention Wide-Ranging Online Data for Epidemiologic Research 2011–2018.** IRRs for HF-related mortality between quintiles of individual county-level factors in fully-adjusted model relative to lowest quintile. Results are representative of four fully-adjusted negative binomial regression models for age (35-64y and 65-84y) and rural-urban subgroups. Data sources as in Fig 3. HF = heart failure; IRR = incidence rate ratio; PCP = primary care physician; y = year.

socioeconomic characteristics and mortality in rural counties [2, 36, 45] and highlight that the observed rural-urban disparities in HF-related mortality are likely a reflection of social and economic differences between rural and urban communities. While rural status was associated with marginally higher HF-related mortality rates in our fully-adjusted model, this is likely due to unmeasured risk factor exposures to such as diet, wealth, and lifetime exposure to cardiovascular risk factors.

## Role of access to care in rural counties in HF-related mortality rates

Our results do not identify a significant contribution of density of PCPs and cardiologists to county-level HF-related mortality. However, there appears to be a modest but significantly lower HF-related mortality rates with greater physician density among young rural adults with minimal to no effect in older rural adults when examined by quintiles (**Fig 4**). Basu et al. recently demonstrated that county-level cardiovascular mortality rates were lower by 3.0 and 4.9 deaths per 100,000 population with greater density of PCPs and cardiologists of 10 per 100,000 county residents, respectively, between 2002 and 2015 [46]. They also found PCP density in rural counties declined twice as much as urban counties between 2002–2015, and nearly 300 counties had no PCPs at all in 2015. An even larger disparity exists in the access to specialists like cardiologists in rural compared with urban areas; among the rural counties included in our analysis, over half had no practicing cardiologists. Interestingly, however, we found that the IRR between rural and urban counties was not significantly different when controlling for physician density continuously compared with the unadjusted model (**Fig 3**). Because changes in physician density act upstream in the causal pathway of HF-related mortality, changes in county-level HF-related mortality may not be manifested until several years after the corresponding change in physician density.

Lack of access to both PCPs and specialists in rural areas may also reflect and be compounded by recent trends in availability of local inpatient hospital care, an important aspect of the management of chronic HF. Over the past decade, rural counties have experienced disproportionate rates of hospital closures, and in the six years following a hospital closure, rural counties experienced an 8.2% decrease in both PCPs and medical specialists [47]. Expansion of Medicaid coverage may be a viable health policy intervention to target HF-related mortality rates as rates of hospital closure have been lower in states that expanded Medicaid in part due to uninsured patients gaining Medicaid coverage, particularly in rural counties [48, 49]. It will be important for future studies to investigate local changes in HF-related mortality rates following these drastic shifts in access to health insurance, physicians, and hospital care, particularly among rural communities.

Telemedicine may also be an effective way to mitigate the large differences in access to care among rural patients and potentially reduce rural-urban disparities in HF prevention and management. This is of particular importance in the context of the current COVID-19 pandemic when telemedicine has become a more common mode of care delivery for chronic disease management. Several different forms of telemedicine interventions have been proposed in the HF setting, including telephone-based support [50–53] and telemonitoring of clinical status (i.e. blood pressure, body weight, and ECG monitoring) [53, 54]. Meta-analyses have

suggested that telemedicine interventions may be effective in HF patients with significant reductions in HF readmissions, HF-related mortality, and all-cause mortality [55, 56]. However, several recent large clinical trials have failed to show a benefit [50, 54]. The use of easily obtained surrogate measures for clinical status in HF patients such as body weight changes may limit the accuracy of clinical assessment and prevent providers from acting early enough in the disease processes to prevent adverse outcomes. More sensitive measures for worsening clinical status or direct measurement of clinical status with wearables or implanted devices may prove useful in future telemedicine trials [57]. Despite these limitations, telemedicine interventions, if optimized, have the potential to change the landscape of HF management specifically for rural populations who otherwise lack access to specialist care.

## Interaction between race and rurality in HF-related mortality

We also demonstrate that recent increases in HF-related mortality are different between race-sex subgroups. Data on the interaction between rural-urban status and race are sparse. One study in the REGARDS cohort demonstrated that Southern, low-income rural Black participants had 85% higher risk for all-cause mortality compared with White participants and over 50% higher risk compared with other Black participants [58]. We found that young Black adults have experienced markedly higher HF-related mortality rates compared with their White peers. Strikingly, between 2011 and 2018, HF-related mortality among young rural Black men increased over 45% (AAMR 35.9 [31.8–40.0] per 100,000 population in 2011 versus 52.3 [47.4–57.3] in 2018) compared with a 33% increase in young rural White men (AAMR 15.6 [14.8–16.4] in 2011 versus 20.8 [19.8–21.7] in 2018). If current trends continue, HF-related AAMR among young rural Black men will double by the year 2023 when compared with 2011.

## Strengths and limitations

Strengths of our study include broad capture of all cardiovascular deaths in which HF was included as a contributing cause from the multiple cause of death files. This is an important distinction as previous studies that have investigated county-level HF-related mortality have measured HF as an underlying cause of death [2]. HF is often erroneously listed on death certificates as the underlying cause of death when the true underlying cause of death is either poorly understood or an alternative cause that subsequently resulted in HF such as coronary artery disease [59, 60]. By specifying HF as a contributing cause, we are better able to capture decedents in whom HF was a significant contributing diagnosis. Additionally, our study utilizes nationwide data from all death certificates in the US over the 8-year study period. This allowed us to accurately characterize trends, geographic variation, and rural-urban disparities on the national level. Finally, we leveraged and integrated multiple large, publicly available data sets to identify county-level factors that may contribute to HF-related mortality.

Our study should also be interpreted in the context of several limitations. The mortality rates and covariate data used in this study were aggregate data on the county level, and individual-level characteristics of decedents were not available. As such, we could not include the age of individual decedents in our regression modeling strategies. Counties with low event rates over the study period were omitted from the cross-sectional county-level IRR analysis due to unreliable mortality rate estimates (13% omitted in the overall county-level analysis, similar to other published studies [16] on county-level HF-related mortality rates). However, all decedents between 2011 and 2018 were included in the analysis of annual trends and rural-urban disparities in HF-related mortality. Furthermore, the purpose of the county-level analysis was to provide an estimate of the association of county-level factors with HF-related mortality.

While systemic differences likely exist between included and excluded counties, the biological contribution of these characteristics to HF-related mortality rates should not differ. Counties spanning large geographical areas may be significantly heterogeneous with mixed rural and urban neighborhoods. Future dedicated analyses of differences in HF-related mortality over smaller geographical areas (i.e. congressional district) may provide additional insight. Counties with low populations may have high variance in AAMRs. However, because counties with less than 20 HF-related mortality events were excluded from the study, the counties at the highest risk for significantly inflated or deflated AAMRs have been excluded. Decedents aged 85 years or older were excluded from the study given concerns related to accuracy of death certificate coding in this age range [32]. While this introduces potential bias, excluding decedents aged 85 years or older likely resulted in lower, conservative estimates of HF-related mortality, but was unlikely to significantly affect annual trends and between-group differences in AAMRs. Mortality rates in this study are based on death certificate data and ICD-10 codes, which is subject to miscoding and does not distinguish between HF phenotypes such as HF with reduced or preserved ejection fraction. Instead, the definition used in this study encompasses a comprehensive burden of overall HF-related mortality. Finally, county-level data on other clinical and behavioral risk factors for HF (e.g. hypertension, smoking, and diet) were not publicly available to be included in this analysis, but represent important factors on the causal pathway to HF.

## Conclusions

Rural adults in the US experience higher rates of HF-related mortality compared with urban adults. County-level factors such as socioeconomic status and risk factor prevalence account for a significant portion of the higher rates of HF-related mortality observed. Our study decomposing the rural-urban disparity in HF-related mortality identifies complex and multi-level factors and suggests that extensive clinical and public health interventions targeting health and economic policy, socioeconomic disparities, access to care, and clinical and behavioral risk factors will be required to reduce this disparity.

## Supporting information

**S1 Checklist. STROBE, strengthening the reporting of observational studies in epidemiology.**
(DOCX)

**S1 Table. County-level factor data sources and years used in primary and sensitivity analyses.** *Most recent publicly available versions of the respective data sources.
(DOCX)

**S2 Table. Crude number of deaths, crude death rate, and percent of deaths for heart failure-related mortality stratified by urban-rural classification and race-sex groups, center for disease control and prevention wide-ranging online data for epidemiologic research 2011–2018.** HF = heart failure; y = years. *Rural-urban status grouped based on the 2013 NCHS Urban-Rural Classification Scheme for Counties. †Expressed as number of deaths per 100,000 residents.
(DOCX)

**S3 Table. Mean county-level characteristics stratified by age and rurality, 2011.** No. = number; Y = years. *Rural-urban status grouped based on the 2013 NCHS Urban-Rural Classification Scheme for Counties. †Percent of residents over age 65 years, percent of female residents, percent of non-Hispanic Black residents, and percent of Hispanic residents according to the

US Census Bureau 2011 Population and Housing Unit Estimates. ‡Percent of residents in poverty, percent of residents unemployed, and median household income according to the US Census Bureau 2018 Small Area Income and Poverty Estimates Program, and percent of residents uninsured (age 18–64) according to 2011 Small Area Health Insurance Estimates Program. §Percent of residents with diabetes and percent of residents with obesity from the 2011 Behavioral Risk Factor Surveillance System. || Primary care physicians and number of cardiologists per 100,000 residents according to the Health Resources and Services Area Health Resources File 2011 and 2010 statistics, respectively.
(DOCX)

**S4 Table. Multivariable negative binomial regression modeling of latest aggregate county-level factors contributing to excess heart failure-related mortality in rural*** **counties, center for disease control and prevention wide-ranging online data for epidemiologic research 2011–2018.** IRR = incidence rate ratio; y = years. *Rural-urban status grouped based on the 2013 NCHS Urban-Rural Classification Scheme for Counties. †Adjusted demographic factors including percent of residents over age 65 years, percent of female residents, percent of non-Hispanic Black residents, and percent of Hispanic residents according to the US Census Bureau 2018 Population and Housing Unit Estimates. ‡Adjusted for socioeconomic factors including percent of residents in poverty, percent of residents unemployed, percent of residents uninsured age 18–64, and median household income according to the US Census Bureau 2018 Small Area Income and Poverty Estimates Program and 2017 Small Area Health Insurance Estimates Program. §Adjusted for clinical characteristics of residents including percent of residents with diabetes and percent of residents with obesity from the 2016 Behavioral Risk Factor Surveillance System. ||Adjusted for clinician density including number of primary care physicians and number of cardiologists per 100,000 residents according to the Health Resources and Services Area Health Resources File 2017 statistics. ¶Adjusted for all covariates in models 2–5.
(DOCX)

**S5 Table. Associations between county-level factors and age-adjusted heart failure-related mortality rates stratified by rural-urban status and age, center for disease control and prevention wide-ranging online data for epidemiologic research 2011–2018.** IRRs for HF-related mortality between quintiles of individual county-level factors in fully-adjusted model relative to lowest quintile. Results are representative of four fully-adjusted negative binomial regression models for age (35-64y and 65-84y) and rural-urban subgroups. Data sources as in Fig 3. HF = heart failure; IRR = incidence rate ratio; y = year.
(DOCX)

## Acknowledgments

The authors take responsibility for decision to submit the manuscript for publication. Dr. Khan had full access to all the data in the study and take responsibility for the integrity of the data and the accuracy of the data analysis.

## Author Contributions

**Conceptualization:** Jacob B. Pierce, Sadiya S. Khan.

**Data curation:** Jacob B. Pierce, Sadiya S. Khan.

**Formal analysis:** Jacob B. Pierce.

**Funding acquisition:** Jacob B. Pierce, Sadiya S. Khan.

**Investigation:** Jacob B. Pierce, Nilay S. Shah, Lucia C. Petito, Lindsay Pool, Donald M. Lloyd-Jones, Joe Feinglass, Sadiya S. Khan.

**Methodology:** Jacob B. Pierce, Lucia C. Petito, Sadiya S. Khan.

**Project administration:** Jacob B. Pierce, Sadiya S. Khan.

**Supervision:** Jacob B. Pierce, Sadiya S. Khan.

**Visualization:** Jacob B. Pierce, Sadiya S. Khan.

**Writing – original draft:** Jacob B. Pierce, Sadiya S. Khan.

**Writing – review & editing:** Jacob B. Pierce, Nilay S. Shah, Lucia C. Petito, Lindsay Pool, Donald M. Lloyd-Jones, Joe Feinglass, Sadiya S. Khan.

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
