## [Decision Letter · Decision Letter 0]

9 Dec 2020

PONE-D-20-34927

Trends in heart failure-related cardiovascular mortality in rural versus urban United States counties, 2011-2018: A cross-sectional study

PLOS ONE

Dear Dr. Khan,

Thank you for submitting your manuscript to PLOS ONE. After careful consideration, we feel that it has merit but does not fully meet PLOS ONE’s publication criteria as it currently stands. Therefore, we invite you to submit a revised version of the manuscript that addresses the points raised during the review process.

Please carefully address each comment by the reviewers.

We look forward to receiving your revised manuscript.

Kind regards,

Jim P Stimpson, PhD

Academic Editor

PLOS ONE

Journal Requirements:

2.Thank you for stating the following financial disclosure:

 [The funders had no role in study design, data collection and analysis, decision to publish, or preparation of the manuscript.].

3.We note that [Figure(s) 2] in your submission contain map images which may be copyrighted. All PLOS content is published under the Creative Commons Attribution License (CC BY 4.0), which means that the manuscript, images, and Supporting Information files will be freely available online, and any third party is permitted to access, download, copy, distribute, and use these materials in any way, even commercially, with proper attribution. For these reasons, we cannot publish previously copyrighted maps or satellite images created using proprietary data, such as Google software (Google Maps, Street View, and Earth). For more information, see our copyright guidelines: http://journals.plos.org/plosone/s/licenses-and-copyright.

1.    You may seek permission from the original copyright holder of Figure(s) [2] to publish the content specifically under the CC BY 4.0 license. 

Reviewers' comments:

Reviewer's Responses to Questions

**Comments to the Author**

1. Is the manuscript technically sound, and do the data support the conclusions?

Reviewer #1: Yes

Reviewer #2: Yes

2. Has the statistical analysis been performed appropriately and rigorously? 

Reviewer #1: Yes

Reviewer #2: Yes

3. Have the authors made all data underlying the findings in their manuscript fully available?

Reviewer #1: Yes

Reviewer #2: Yes

4. Is the manuscript presented in an intelligible fashion and written in standard English?

Reviewer #1: Yes

Reviewer #2: Yes

5. Review Comments to the Author

Reviewer #1: The manuscript is well written and deals with an area of major public health significance, there appropriately and well to a set of 3 reviewers. I would add the following comments, mostly relating to tables:

Table S2. Suggest that the entries in the “crude rate” columns all be shown to one place after the decimal point (including the whole numbers like “92” for the crude mortality rate of Black women.

Table S3. I could not understand the entries in the row for “County population, No.”, e.g what do “918,685” and “(1,559,985)” represent?

Figure 4. The scale of this figure is so small that is difficult to visually determine whether some of the lines cross “1” x-axis point. Because of this, I suggest that an additional table be created with the values shown in the graph.

Of broader and more substantive concern is the absence of consideration of the major behavior risk factors and of hypertension in these data (recognizing that reasonable data may not exist) or in the limitations section of the discussion. The specific behaviors are smoking, diet, and physical activity. Diet and physical activity may be along the causal pathway to obesity and diabetes, which show rural-urban differences in the manuscript. Smoking is a known risk factor for heart failure and is more prevalent in rural vs. urban communities. Hypertension is the major risk factor for heart failure and is also more prevalent in rural the urban areas in the U.S. I would section that a not too long section be added in the discuss for this which might just be that there were other factors (e.g., the ones noted in this paragraph) for which data were not available to include in the analysis

Reviewer #2: We thank the authors for the opportunity to rereview the manuscript entitled “Trends in heart failure-related cardiovascular mortality in rural vs urban United States counties, 2011-2018: a cross-sectional study”. The authors have responded appropriately to the prior queries. However, I have one additional concern that I would ask the authors to respond to.

1. On pg 12 line 241, the authors note that “There were no practicing cardiologists in 51.1% and 74.9% of rural counties”. Do the authors have a way of knowing whether patients are referred to practicing cardiologists who might be located in urban (or other) locations) for these rural patients? Similarly, how does the adjustment for physician density change if the authors adjust for the number of PCPS and the number of cardiologists in separate models ??

6. PLOS authors have the option to publish the peer review history of their article (what does this mean?). If published, this will include your full peer review and any attached files.

Reviewer #1: No

Reviewer #2: No

---

## [Author Response · Author response to Decision Letter 0]

20 Jan 2021

Editor Comments: 

Comment #1: Please ensure that your manuscript meets PLOS ONE's style requirements, including those for file naming. The PLOS ONE style templates can be found at

Author Reply: We have configured our manuscript to PLOS ONE’s style requirements, specifically non-bold title, removing degrees from author list, correction of section title font sizes, and correction of sentence case for section titles. 

Comment #2: Thank you for stating the following financial disclosure:

 [The funders had no role in study design, data collection and analysis, decision to publish, or preparation of the manuscript.].

- Please clarify the sources of funding (financial or material support) for your study. List the grants or organizations that supported your study, including funding received from your institution.

- State what role the funders took in the study. If the funders had no role in your study, please state: “The funders had no role in study design, data collection and analysis, decision to publish, or preparation of the manuscript.”

- If any authors received a salary from any of your funders, please state which authors and which funders.

- If you did not receive any funding for this study, please state: “The authors received no specific funding for this work.”

Author Reply: We have included the following statement in the cover letter.

This work was supported by grants from National Institutes of Health's National Center for Advancing Translational Sciences Grant Number KL2TR001424 and the American Heart Association (#19TPA34890060) to SSK including salary support. The funders had no role in study design, data collection and analysis, decision to publish, or preparation of the manuscript.

Comment #3: We note that [Figure(s) 2] in your submission contain map images which may be copyrighted. All PLOS content is published under the Creative Commons Attribution License (CC BY 4.0), which means that the manuscript, images, and Supporting Information files will be freely available online, and any third party is permitted to access, download, copy, distribute, and use these materials in any way, even commercially, with proper attribution. For these reasons, we cannot publish previously copyrighted maps or satellite images created using proprietary data, such as Google software (Google Maps, Street View, and Earth). For more information, see our copyright guidelines: http://journals.plos.org/plosone/s/licenses-and-copyright.

Author Reply: Thank you for this query. We confirm that Figure 2 in our manuscript is an original figure that we produced using the ArcGIS software and raw data from CDC WONDER as detailed in the methods section of the manuscript. We have included an attribution for the basemap layer used in ArcGIS to create the figure in the caption of Figure 2 as recommended under the Creative Commens Attribution License (CC BY 4.0). The caption now includes the following attribution:

Page 12, lines 254-256: “Basemap Sources: Esri, TomTom North America, Inc., U.S. Census Bureau, U.S. Department of Agriculture (USDA), National Agricultural Statistics Service (NASS)”

Reviewer #1: The manuscript is well written and deals with an area of major public health significance, there appropriately and well to a set of 3 reviewers. I would add the following comments, mostly relating to tables.

Author Reply: We thank the reviewer for their support of our work, and we have responded to each comment below. 

Table S2. Suggest that the entries in the “crude rate” columns all be shown to one place after the decimal point (including the whole numbers like “92” for the crude mortality rate of Black women.

Author Reply: We have made this change as the reviewer suggested to Table S2.

Table S3. I could not understand the entries in the row for “County population, No.”, e.g what do “918,685” and “(1,559,985)” represent?

Author Reply: We thank the reviewer for the clarifying point and have revised the title and table to clarify that the table reflects the mean values for each of the county-level characteristics included in each age-group analysis. County population is reflected in number (No.) and the cell parenthetically denotes the standard deviation. The title has been revised to: “S3 Table. Mean county-level characteristics stratified by age and rurality, 2011” and the row title has been revised to “County population, No. (SD).”

Figure 4. The scale of this figure is so small that is difficult to visually determine whether some of the lines cross “1” x-axis point. Because of this, I suggest that an additional table be created with the values shown in the graph.

Author Reply: We have included S5 Table, which details the IRR point estimates and 95% CI displayed graphically in Fig 4. The caption for the new table reads:

• Page 29, lines 648-654: “S5 Table. Associations between county-level factors and age-adjusted heart failure-related mortality rates stratified by rural-urban status and age, Center for Disease Control and Prevention Wide-Ranging Online Data for Epidemiologic Research 2011-2018. IRRs for HF-related mortality between quintiles of individual county-level factors in fully-adjusted model relative to lowest quintile. Results are representative of four fully-adjusted negative binomial regression models for age (35-64y and 65-84y) and rural-urban subgroups. Data sources as in Fig 3. HF = heart failure; IRR = incidence rate ratio; y = year.”

Of broader and more substantive concern is the absence of consideration of the major behavior risk factors and of hypertension in these data (recognizing that reasonable data may not exist) or in the limitations section of the discussion. The specific behaviors are smoking, diet, and physical activity. Diet and physical activity may be along the causal pathway to obesity and diabetes, which show rural-urban differences in the manuscript. Smoking is a known risk factor for heart failure and is more prevalent in rural vs. urban communities. Hypertension is the major risk factor for heart failure and is also more prevalent in rural the urban areas in the U.S. I would section that a not too long section be added in the discuss for this which might just be that there were other factors (e.g., the ones noted in this paragraph) for which data were not available to include in the analysis

Author Reply: We agree that county-level data on these factors would have been interesting to include in our analysis. As the reviewer suggested, we have included a brief discussion of this in the limitations section. The text now reads:

• Page 19, lines 431-433: “Finally, county-level data on other clinical and behavioral risk factors for HF (e.g. hypertension, smoking, and diet) were not publicly available to be included in this analysis, but represent important factors on the causal pathway to HF.”

Reviewer #2: We thank the authors for the opportunity to rereview the manuscript entitled “Trends in heart failure-related cardiovascular mortality in rural vs urban United States counties, 2011-2018: a cross-sectional study”. The authors have responded appropriately to the prior queries. However, I have one additional concern that I would ask the authors to respond to.

Author Reply: We thank the reviewer for their support of our work, and we have responded to the reviewer request below. 

Comment #1: On pg 12 line 241, the authors note that “There were no practicing cardiologists in 51.1% and 74.9% of rural counties”. Do the authors have a way of knowing whether patients are referred to practicing cardiologists who might be located in urban (or other) locations) for these rural patients? Similarly, how does the adjustment for physician density change if the authors adjust for the number of PCPS and the number of cardiologists in separate models ??

Author Reply: We thank the reviewer for raising this important question. However, data are not available to evaluate the location of the cardiologist to which patients are being referred. Regarding the adjustment for number of cardiologists and PCPs separately, we tested this as suggested by the reviewer, and the results are shown below. We observed only marginal differences in the rural-urban disparity in HF-related mortality when compared with the adjustment for the two variables together (Table below).

 IRR for rural vs. urban HF-related mortality (95% CI)

Model (Clinicians per 100,000 residents) Age 35-64 y Age 65-84 y

PCPs and Cardiologists 1.57 (1.47, 1.67) 1.14 (1.11, 1.16)

PCPs alone 1.53 (1.44, 1.63) 1.13 (1.11, 1.15)

Cardiologists alone 1.57 (1.47, 1.67) 1.13 (1.10, 1.15)

---

## [Editor Report · Decision Letter 1]

27 Jan 2021

Trends in heart failure-related cardiovascular mortality in rural versus urban United States counties, 2011-2018: A cross-sectional study

PONE-D-20-34927R1

Dear Dr. Khan,

We’re pleased to inform you that your manuscript has been judged scientifically suitable for publication and will be formally accepted for publication once it meets all outstanding technical requirements.

Kind regards,

Jim P Stimpson, PhD

Academic Editor

PLOS ONE

---

## [Editor Report · Acceptance letter]

9 Feb 2021

PONE-D-20-34927R1 

Trends in heart failure-related cardiovascular mortality in rural versus urban United States counties, 2011-2018: A cross-sectional study 

Dear Dr. Khan:

I'm pleased to inform you that your manuscript has been deemed suitable for publication in PLOS ONE. Congratulations! Your manuscript is now with our production department. 

Kind regards, 

on behalf of

Dr. Jim P Stimpson 

Academic Editor

PLOS ONE